evolution, ecology

interspecific competition, ecomorphological trait evolution, avian diversification

**Author for correspondence:**
A. M. Chira
e-mail: angelamchira@gmail.com

# The signature of competition in ecomorphological traits across the avian radiation

A. M. Chira[1,2], C. R. Cooney[1], J. A. Bright[3], E. J. R. Capp[1], E. C. Hughes[1], C. J. A. Moody[1], L. O. Nouri[1], Z. K. Varley[1] and G. H. Thomas[1,4]

[1]Department of Animal and Plant Sciences, University of Sheffield, Sheffield S10 2TN, UK
[2]Department of Biology, Washington University in St Louis, St Louis, MO, USA
[3]Department of Biological and Marine Sciences, University of Hull, Hull, UK
[4]Bird Group, Department of Life Sciences, The Natural History Museum, Tring, Hertfordshire, UK

(iD) AMC, 0000-0002-2964-7583; CRC, 0000-0002-4872-9146; GHT, 0000-0002-1982-6051

Competition for shared resources represents a fundamental driver of biological diversity. However, the tempo and mode of phenotypic evolution in deep-time has been predominantly investigated using trait evolutionary models which assume that lineages evolve independently from each other. Consequently, the role of species interactions in driving macroevolutionary dynamics remains poorly understood. Here, we quantify the prevalence for signatures of competition between related species in the evolution of ecomorphological traits across the bird radiation. We find that mechanistic trait models accounting for the effect of species interactions on phenotypic divergence provide the best fit for the data on at least one trait axis in 27 out of 59 clades ranging between 21 and 195 species. Where it occurs, the signature of competition generally coincides with positive species diversity-dependence, driven by the accumulation of lineages with similar ecologies, and we find scarce evidence for trait-dependent or negative diversity-dependent phenotypic evolution. Overall, our results suggest that the footprint of interspecific competition is often eroded in long-term patterns of phenotypic diversification, and that other selection pressures may predominantly shape ecomorphological diversity among extant species at macroevolutionary scales.

## 1. Background

A fundamental topic of interest in evolutionary biology is understanding the tempo and mode underlying the accumulation of morphological diversity at macroevolutionary scales [1]. The most commonly used statistical models (e.g. Brownian motion (BM) and Ornstein–Uhlenbeck (OU)) for modelling phenotypic differentiation among species assume that lineages evolve independently from each other. However, evolution in the absence of biotic interactions between related lineages may be unrealistic in many (if not all) cases, and competition for shared resources in particular has been closely linked with patterns of trait diversification [2–4]. Intuitively, and all else being equal, the absence of competitors offers species access to a multitude of free niches, which can result in bursts of phenotypic diversification, as shown in iconic island radiations [5,6]. Alternatively, the presence of competitors can result in rapid phenotypic differentiation as lineages become more specialized to avoid competition [7,8] or cause convergence in traits involved in competitor recognition [9]. Furthermore, the presence or accumulation of many competitors can limit trait evolution if niche spaces are bounded [10,11] and even cause the extinction of competitors [12]. While competition often emerges as a powerful selective force in studies involving a small number of taxa (e.g. [13,14]), testing the strength of competitive selection pressures at macroevolutionary scales has been hindered by the difficulty of incorporating species interactions into models of trait evolution, often combined

with a lack of well-resolved trait and phylogenetic data [15,16]. Hence, we do not know the importance of competition beyond several intensively studied radiations [5,14] and thus more generally whether biotic interactions impact the accumulation of phenotypic diversity across global radiations.

Competitive interactions between related species have been linked to several patterns of trait evolution. Diversity-dependent models, in which the rate of evolution changes as a function of the number of accumulating lineages [10], are often used as evidence for the impact of species interactions on evolutionary rates within clades. In particular, the pattern of decreasing evolutionary rates with an increasing number of related species is interpreted as a signal of reduced potential for further evolution in niches saturated by competitors [11] but see [17]. Conversely, competitive interactions have also been linked with fast trait diversification, as they represent a dynamic selection pressure, potentially leading to rapid changes in species' phenotypes via character displacement [18]. The expected signature of character displacement is represented by highly structured covariances among species' phenotypes (i.e. throughout the evolutionary history of a group, the covariance between the phenotypes of clade members reflects which species compete for the same resources and if all species in a clade compete then trait histories match closely to phylogenetic structure) and, overall, overdispersed trait distributions among extant species.

Recently, more mechanistic models of trait evolution have been developed, which incorporate the effect of competitive selection pressures between related lineages on phenotypic diversification [19,20]. These novel methods are based on the equation of a random walk model of evolution [21], to which terms that account for competition are added. The underlying assumption is that similarity in relevant traits (i.e. traits involved in the acquisition of limiting resources) between species corresponds to stronger competition. Thus, these models aim to detect patterns of exaggerated trait divergence between closely related lineages, as expected if species differentiate to avoid costly competitive interactions [5,14]. Species can evolve away from the mean trait value among congeners [19], or more subtly, the amount of morphological divergence can be modelled for all pairs of species in a clade as a function of pairwise similarity in target traits [20]. By contrasting the fit of trait evolutionary models that assume the presence of competition with models in which species evolve independently from each other, we can determine the relative importance of various hypotheses in shaping phenotypic evolution within clades. These methods have already been applied to a few radiations [20,22], but we lack a comprehensive multi-clade and deep-time perspective on just how often ecological processes like competition leave a signature in the dynamics of phenotypic accumulation.

Here, we investigate the prevalence of signatures of competition in the evolution of avian ecomorphological traits. We focus on avian beak shape and size, as well as body mass—traits which are closely related to resource acquisition (e.g. [23,24]). We contrast methods that look for trait-dependent (species within clades evolve to increase morphological differentiation among each other) or diversity-dependent evolution (rates of trait evolution are dependent on the accumulation of clade members) to models of evolution in which species evolve independently from their relatives to describe the prevalence of evolutionary patterns consistent with a signature of species interactions in avian clades.

We describe trait evolutionary trajectories in well-defined avian orders and super-families, as well as in more recent radiations within these groups, to obtain a comprehensive perspective of the signal of interactions between related species across Aves.

## 2. Methods

### (a) Morphological data

We used three-dimensional scans of bird beaks to collect beak shape and size measurements for 8748 avian species. A detailed account of the protocols used to extract these data is given in [25]. Briefly, we used study skins from the Natural History Museum in Tring and Manchester Museum and scanned the beaks using white- and blue-structured light scanning (*Flex-Scan3D*, LMI Technologies, Vancouver, Canada). We further extracted information about variation in shape and size between scans using landmark-based geometric morphometrics [26]. We chose four key points on the avian beak, which can be easily placed repeatedly across specimens: (i) the tip of the beak, (ii) the posterior margin of the beak on the dorsal midline, (iii) the left, and (iv) the right tomial edges. Furthermore, we had 75 semi-landmarks that unite (i) to (ii), (iii) and (iv), forming the dorsal midline, and left and right tomial edges, respectively. The authors and members of the public used the crowdsourcing website http://www.markmybird.org to complete the landmarking process. Each three-dimensional image was marked by at least three independent users, and unsuitable landmark efforts (either poor landmarking of individual scans or non-similarity in the landmark position between users) were discarded following quality control protocols (see [25,27]).

Landmark configurations were subjected to a generalized Procrustes analysis (to remove the effects of any geometric information unrelated to shape) and alignment [28]. We then used principal component (PC) and phylogenetic PC (pPC) analyses [29,30] on user-averaged landmarks to extract the main axes of beak shape variation. We focus on the first four PC axes for beak shape (accounting a total of 95% variation, electronic supplementary material, figure S1a), relative beak size and body mass in the main text and report the results from the pPC analyses in the electronic supplementary material. We used the square root of the sum of squared distances (i.e. the Euclidean distance) of landmarks from their centroid, i.e. centroid size [26] as a measure of beak size, and we extracted species body mass measurements from the EltonTraits database [31]. We separately modelled the evolution of body mass, beak size and beak shape. Finally, we regressed beak size against body mass using phylogenetic generalized least squares regressions [32,33] within individual clades and used the residuals as a proxy for beak size relative to body size in evolutionary models. Measurement error could not be included as the data comprised of one individual per species.

### (b) Phylogenetic data

We split species into well-supported orders and super-families, as identified by Jetz *et al.* [34]. We used 10 000 trees from the posterior distribution of 'full' trees (i.e. trees with all 9993 avian species) and 'genetic' trees (i.e. trees including only species for which genetic data were available) accessed from http://www.birdtree.org [34]. We pruned these trees to obtain a posterior distribution of phylogenies for each clade and further used TreeAnnotator [35] to generate maximum clade credibility (MCC) trees, setting branch lengths equal to common ancestor node heights. These consensus trees were further pruned to species for which we have beak shape and size data. For swifts and treeswifts, owls, tapaculos and antpittas, the percentage of species with

morphological data ranged between 30 and 57%, and for all other clades, these percentages were higher than 70%. We only considered groups with at least 20 species, to avoid potential flawed model fit owing to small clade sizes (clade size ranged from 21 to 195 species) [19]. In addition, we generated finer scale clades by time-slicing each of the clades defined above at the mid-point of the root-tip distance in each individual clade and extracting the subclades with at least 20 members that were younger than the time slice. These subclades essentially represent recent radiations within each parent clade, and we used them to test for the prevalence of competition signal at finer macroevolutionary scales. The electronic supplementary material, appendix S1 provides a list of all species used, their division into clades, as well as the associated beak size, shape and body mass data.

## (c) Models of trait evolution

We used a suite of evolutionary models that assume species evolve independently from their close relatives to test how trait diversity accumulates across avian clades. Specifically, we fit models of (i) random walk trait divergence (BM [21]), (ii) divergence with a restraining parameter (OU [32,36]) and (iii) time-dependent models, in which the rate of evolution changes in time [37,38]. We contrasted the fit of these models with two types of models that are considered to be consistent with different outcomes of interspecific competition. These were diversity-dependent models, in which the rate of evolution varies (iv) linearly, or (v) exponentially with the number of lineages [10], and (vi) the recently developed matching competition model (MC [19,22]), in which key traits evolve away from the mean values of clade members. Both positive and negative diversity-dependence could be considered as signals of interspecific competition. However, within each clade not all species will compete for the same resources, and the signal of positive diversity-dependence should be structured according to which species are in competition. Not taking into account the structured trait covariances resulting from a process of character displacement could result in the misidentification of a signal for competitive species interactions within clades with a scenario in which rates of trait evolution increase with the accumulation of any species. We modelled this by applying diversity-dependent and MC models that also take into account the ecological properties of clade members [39]. Specifically, for each clade, we restricted interactions between clade members to species that share the same diet category and foraging strategies. We followed Felice et al. [40] and used the EltonTraits database to split species' diets into: (i) invertebrates, (ii) terrestrial vertebrates, (iii) fishes, (iv) carrion, (v) fruit, (vi) nectar, (vii) seeds and (viii) plants. We also used EltonTraits and defined forage categories as: (i) ground, (ii) understory, (iii) mid-high, (iv) canopy feeders, (v) aerial foragers, and further, pelagic foragers (vi) below and (vii) above the water surface, and non-pelagic foragers (ix) below and (x) above the water surface. For each clade, we used these values in a PC analysis (PCA) for categorical data. We used the first two PCs to cluster species into four ecoguilds correspondent with the four quadrants of the plot regressing PC1 against PC2 (electronic supplementary material, appendix S1 and figure S1b). Thus, species that are similar in their foraging strategy and diet category grouped in the same ecoguild. We used make.simmap [41] to build 50 stochastic maps that represent an estimation of species' ecoguild membership throughout the history of the clade [39]. Only species that share the same ecoguild are allowed to interact in diversity-dependent and matching competition models of trait evolution. By using a PCA, we reduced the potential large number of diet and foraging categories to four ecoguilds, decreasing the possibility of stochastic maps inferring large (and probably unreliable) numbers of shifts. Clades in which ecoguild reconstruction and/or subsequent

model fitting produced suspect reconstructions (e.g. when several branches showed patterns of many transitions backwards and forwards between ecoguild on the same edge) were excluded from the analyses (18 groups among orders and super-families, and seven groups among the more finely divided clades, electronic supplementary material, appendix S1).

In total, we summarize patterns of trait evolution for 59 avian orders and super-families (total species = 5108), as well as 93 clades obtained from a finer division of clades (total species = 4637). We also repeated the analyses using trees for which genetic data are available for all species in the Jetz et al. [34] tree (54 clades), to ensure that our results are robust to potential phylogenetic error associated with including branches with non-genetic data in the trees (results in the electronic supplementary material). Additionally, we quantified distances between the consensus trees and the posterior distribution of trees to test whether the prevalence of competition signal correlates with the degree of phylogenetic uncertainty (see the electronic supplementary material for results). While these analyses cannot rule out variation in model inference associated with different tree topologies, they do allow us to assess possible biases towards certain models when phylogeny is uncertain.

Models with competition also allow for the incorporation of stochastic maps that represent the reconstructed geographical relationships between species. This approach is recommended for clades with simple biogeography histories (e.g. Anolis lizards across the Greater Antilles [19]). While defining the biogeographical context of many avian clades is probably much more complex, stochastic map reconstructions can be informative, and so we performed an additional set of analyses incorporating the biogeographical history of clades rather than ecoguild membership. We did not combine the geographical and ecoguild structures, as this can lead to a very high number of discrete character states per clade. These methods and results of these analyses are described in detail in the electronic supplementary material.

## (d) False positives in the signal for interspecific competition

A common concern about comparative methods is that many processes can give rise to the same distribution of traits at the tip of the phylogeny [42]. Under the Akaike information criterion (AIC), models with competition have been shown to be distinguishable from models which assume that species evolve independently from each other when competition is present in the generating model [19,22]. However, AICc-based model selection can favour diversity-dependent models in which rates increase towards the present when the true evolutionary model is a BM process in the presence of noise or an OU process (e.g. [19]). We addressed the issue of false positives in diversity-dependent patterns by using simulated datasets. Specifically, we chose avian phylogenies of small (25 species), intermediate (51 and 112 species) and large sizes (195 species) from the pool of empirical avian trees. We simulate OU processes of trait evolution, setting the rate of evolution (sigma parameter) to 0.1, and two possible values for the strength of constraints ($\alpha$ parameter): $\alpha = 1$ and $\alpha = 5$, respectively. We standardized trees to a depth of 1 prior to simulations. Each process was simulated 1000 times on each tree. We applied diversity-dependent and the MC models on these simulated trees and datasets, and quantified the proportion of false positives for each model (i.e. the frequency at which each model with competition is chosen over the OU model at an AICc difference greater than two units). We considered models in which all species can interact, as well as models where the pool of interactive species is restricted to close relatives that share the same ecoguild.

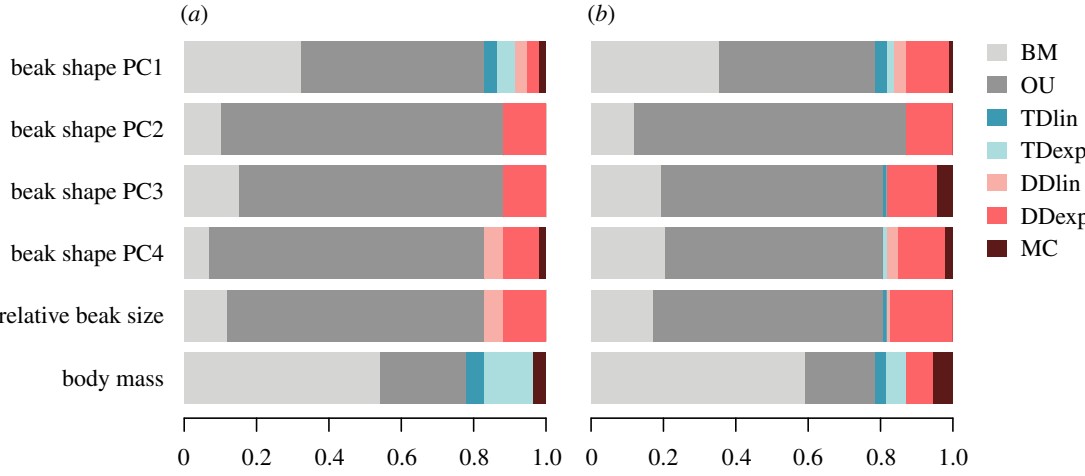

**Figure 1.** Model support (the proportion of times each model is chosen as best, i.e. smallest AICc values) across clades when modelling the evolution of various ecomorphological traits. We consider models without competition: Brownian motion (BM), Ornstein–Uhlenbeck (OU), linear (TDlin) and exponential (TDexp) time-dependent, and models with competition: linear (DDlin) and exponential (DDexp) diversity-dependent, and the matching competition (MC). Results summarized across (a) avian orders and super-families, 59 clades, and (b) finer macroevolutionary scales, 93 clades. (Online version in colour).

### (e) Phylogenetic signal and the effect of clade age on the signature of competition across clades

We built a MCC tree for all 8748 species using 1000 trees from the posterior distribution of 'full' trees and then collapsed branches representing species from the same clade (i.e. each tip label represents a monophyletic clade, rather than individual lineages). We used the D statistic for binary traits [43] and this MCC tree to determine the degree of phylogenetic clustering in the signal of competitive interactions. Each clade was given a score of 0 or 1 (absence/presence of competition signal) depending on whether the best supported model was a diversity- or trait-dependent model, with an AICc difference greater than two compared with any model that assumed lineages evolved independently. Furthermore, we expect that competition, if prevalent, will be more readily detectable in younger clades, as the footprint of inter-specific interactions is less likely to have been eroded by much broader-scale patters. We used a phylogenetic logistic regression [44,45] to test the correlation between the signal of competition and the age of the most recent common ancestor of clade members. We included clade size as a covariate to distinguish between the effect of clade age versus potential-biased selection for models with competition in smaller clades (if younger clades are also smaller). We used the MCC tree for all 8748 species to estimate the crown age of each clade and the trees with collapsed branches to account for the phylogenetic relatedness between clades in the logistic regression. All analyses were performed using R (notably, RPANDA [46], caper [47], geiger [48], phylolm [45] and phytools [41]).

## 3. Results

### (a) The fit of trait evolutionary models across avian clades

We find that the BM and OU models are the best-fitting models (smallest AICc) in the majority of orders and super-families across all the ecomorphological traits considered (figure 1a; electronic supplementary material, table S1a). In comparison, the time-dependent, diversity-dependent and MC models are not commonly the best-fitting models for any clade or trait. When best supported by the data, diversity-dependent models suggest an increase in evolutionary

rates with an increasing number of species (electronic supplementary material, figure S2). We find substantial support for models with competition (i.e. smallest AICc with $\Delta AiCc > 2$ from the best model assuming lineages evolve independently) in 14 out of 59 clades for beak shape (i.e. either across PC1, PC2, PC3 or PC4), seven for relative beak size and only one for body mass. Two clades show support for these models in more than one trait (beak shape and beak size), and overall, 20 out of 59 clades show strong support for diversity-dependent (19 clades, out of which 17 clades show positive diversity-dependence) or MC (two clades) models in either beak shape, size or body mass (figure 2). In several clades, models with competition are the best fit of the data from one or multiple models assuming independent evolution from related lineages. The electronic supplementary material, appendix S2 provides estimated model parameters and model fit for each clade. The analyses including the biogeographical context show similar patterns (electronic supplementary material, appendix S3, figures S2d and S3, and table S1b). In many clades, purely geographical analyses miss the signal for competition otherwise revealed by taking into account species' ecoguild membership. However, these analyses also uncover several cases where the signature of competition had been missed. When considering the combined effect of both geography and ecology, i.e. a clade is considered as showing a competition signal in either of these analyses, 27 out of 59 clades show strong support for competition in beak shape, size or body mass.

We further summarized patterns of trait evolution across more finely divided avian groups by replacing the bigger super-families with more recently radiated monophyletic subclades. We again find that models such as the BM or the OU best explain the data in the vast majority of clades (figure 1b; electronic supplementary material, table S1c), but we note a 6% increase in the frequency for a signal of competition in clades (electronic supplementary material, figure S4). Out of a total of 93 clades, we find strong support for models with competition in 25 clades for beak shape, 10 for relative beak size and five for body mass. Overall, 37 out of 93 clades show a pattern of evolution consistent with a signal of species interactions (33 clades for diversity-dependence

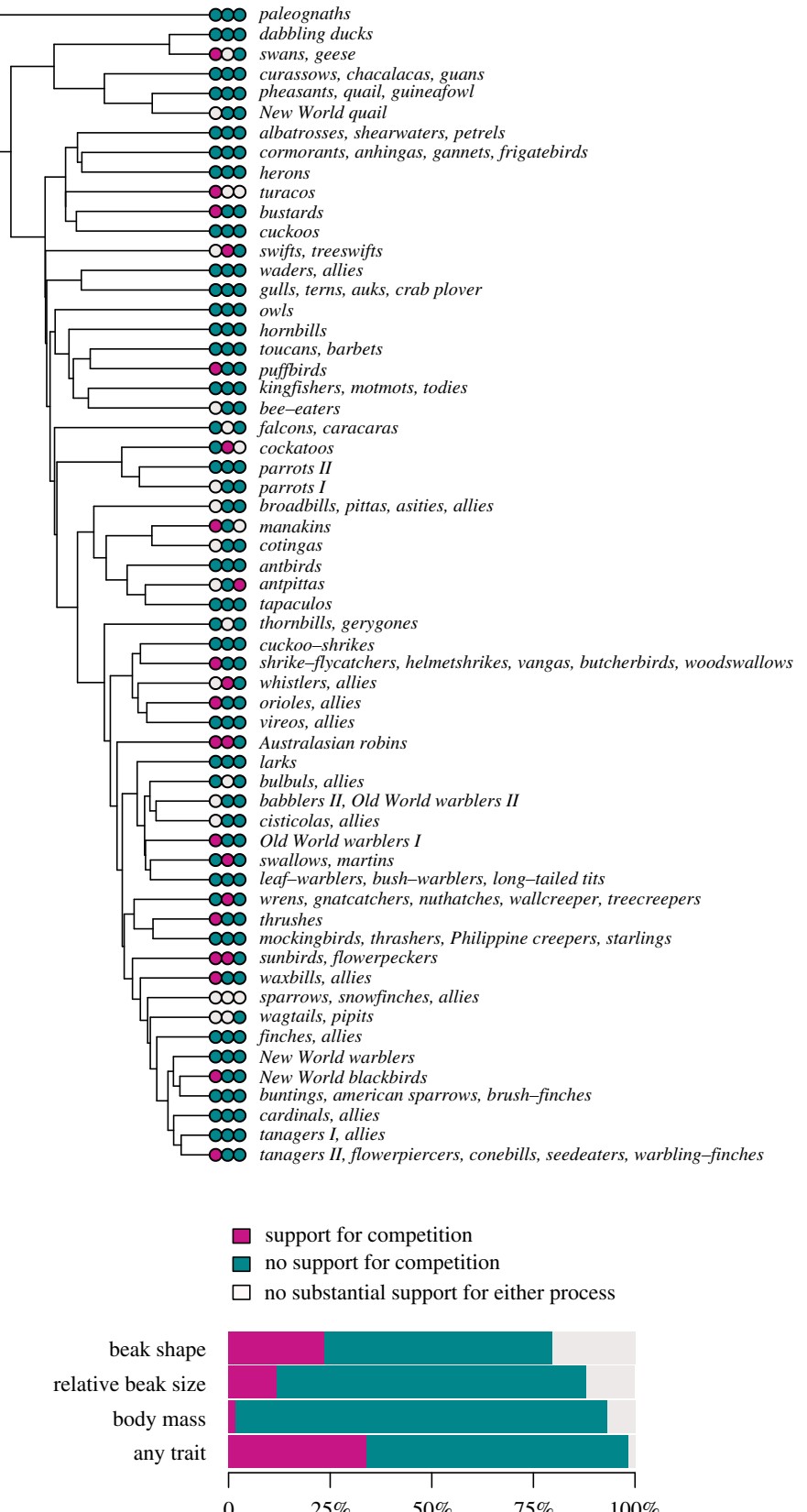

**Figure 2.** The signature of competition across 59 avian orders and super-families. Support for competition (purple) is determined by whether a diversity- or trait-dependent model best explains the data in the focal clade, with an AICc difference greater than two compared with any model assuming lineages evolve independently. The following ecomorphological traits are considered (left to right circles): beak shape (purple indicates the signal of competition in either PC1, PC2, PC3 or PC4), relative beak size and body mass. Twenty out of 59 clades show support for competition in either beak shape, size or body mass ('any trait' in the stacked bars). Clades where models with and without competition cannot be distinguished by an AICc difference greater than two are marked by light grey. (Online version in colour).

and 6 for the MC) in either beak shape, size or body mass (electronic supplementary material, figure S4). The electronic supplementary material, appendix S4 provides estimated model parameters and model fit for each clade.

## (b) False positives in the fit of models with interactions

The frequency at which models with competition are chosen over the OU model with an AICc difference greater than two is small (less than 5%) when the trait data are simulated with

a small strength of constraints ($\alpha = 1$, electronic supplementary material, figures S5a and S6). When the strength of constraints is high ($\alpha = 5$), we find that the frequency of false positives for exponentially diversity-dependent models is high in small and intermediate clade sizes (frequencies of up to 40%, electronic supplementary material, figure S7). Conversely, applying models that take into account the ecological guild membership of species (and hence the expected structured covariance in phenotypes) alleviates the errors in model selection. False positive rates for models with ecoguild were generally smaller than 5% (electronic supplementary material, figures S5–S7). Yet, we found false positive support for diversity-dependent models over the true OU process of evolution in 14% of the smallest clades (25 species) when the strength of constraints is high ($\alpha = 5$, electronic supplementary material, figure S7). This suggests that caution is necessary when interpreting support for the exponentially increasing diversity-dependent models when (i) potential interactions are not accounted for and (ii) trees are small. Lastly, we find that when an OU process of evolution is confused with models featuring competition, the signal in such models is predominantly of positive diversity-dependence (i.e. an estimated increase in the rate of evolution with the number of species) or of negative trait-dependent divergence (i.e. a MC scenario with negative $S$ parameter, suggesting that traits evolve away from the mean of the clade, electronic supplementary material, figures S5–S7).

## (c) Phylogenetic signal and the effect of clade age on the signature of competition across clades

We find no evidence for phylogenetic signal in the support for competition across avian clades. $D$ statistic values ($D = 1.463$ for orders and super-families, and $D = 0.808$ for more finely divided clades) are not different from expectations under a random phylogenetic structure when looking across any eco-morphological trait (electronic supplementary material, table S2). Our results thus show that variation in the signal for competition is dispersed randomly with respect to phylogeny across the avian tree. Furthermore, we find no effect of clade age on the overall signal of competitive interactions (i.e. a signature across either beak shape, size or body mass) either in avian orders and super-families, or at finer macroevolutionary scales (electronic supplementary material, tables S3 and S4).

We acknowledge that the use of phylogenetic and standard PC axes of variation in univariate trait evolutionary models has been criticized [49], and using only the first PC axes can bias model selection criteria towards early burst methods. However, the PCs used here explained a large proportion of beak shape variation (95%), and further, we find little support for early burst models in our dataset. Importantly, the results presented here are mirrored in the analyses using genetic data only trees (electronic supplementary material, figures S2c and S8, tables S4, S5 and S6, and appendix S5), as well as when using phylogenetic PCs for quantifying avian beak shape (electronic supplementary material figure S9, table S7 and appendix S6). We also find no association between a signal of competition and high levels of dissimilarities between the consensus tree and the posterior (electronic supplementary material, table S8). Our findings are thus robust to using trees that include species for which there are no genetic data, and to alternative methods for quantifying the ecomorphological phenotype of species.

## 4. Discussion

Here, we investigate the prevalence for a signature of competitive selection pressures between clade members in the evolution of key ecomorphological traits across the bird radiation. We find that most often the evolutionary trajectories of ecomorphological traits are best explained by models that assume species evolve independently. When looking across any ecomorphological trait, we find support for a signature of competition in 34% of bird orders and super-families (45% when including the combined results across both geography and ecoguild analyses) and 40% for more finely divided clades. Where a signal of competition within clades does occur, we find that models with competition are the best fit for the data across one, rather than multiple axes of phenotypic divergence. Furthermore, we find that the signal of competition largely coincides with positive diversity-dependent evolution, in which the accumulation of closely related species with similar diets and foraging strategies increases trait evolutionary rates. Taken together, our results suggest that competitive selection pressures from related lineages can influence trait evolution in an appreciable number of clades, but that the footprint of such processes on the extant phenotypic diversity is smaller compared with other selection pressures, and/or the signature of biotic interactions is hard to detect in the trait distribution of extant lineages using standard models of trait evolution.

We generally find that a signal of competitive interactions coincides with a strong support for diversity-dependent models that show an increase in rates of ecomorphological evolution with the packing of species. Our results thus indicate that interactions among related lineages could maintain rapid phenotypic changes even with the filling of niches [17,18]. The MC model rarely provides the best description of the data. However, a better fit of positive exponential diversity-dependent models over the MC model has been linked with a signal of both competition and bounded trait evolution occurring simultaneously [22]. Competition drives the partitioning of a certain trait space between interacting species, also leaving a strong phylogenetic structure. With the addition of more species, the increasingly changing adaptive landscape can maintain increasingly fast evolutionary rates, but the phylogenetic structure of the process can be eventually eroded in the presence of a bounded phenotypic space. We also find a comparatively low prevalence for negative diversity-dependence (examples of such patterns included the iconic adaptive radiations in the Malagasy vangas, sunbirds and flowerpeckers). However, in clades where the MC model is best fitting, diversity-dependent models generally show a decrease in evolutionary rates with more species (electronic supplementary material, appendices S2–S6). Thus, while we find relatively scarce evidence for a scenario of competition limiting the potential for evolution once niches become filled and ecological opportunity decreases, our results imply that character displacement and constraint can arise together [50]. We note that in clades where divergence under constraints (when species' traits are pulled towards the mean of the clade) occurs alongside a process of divergence away from the mean trait values of clade members (e.g. under competition), OU models are favoured to models of competition [22]. Our results are thus probably blind to such phenomena. Furthermore, our results are based on extant lineages only. The parameters of the MC model are generally robust to

extinction (as inferences are based on the mean trait values of clade members [19]). However, diversity-dependent models show underestimated slope values with increasing extinction levels [19]. Overall, these results are consistent with a recent simulation study exploring trait evolutionary trajectories expected under competitive selection pressures [17], which revealed that cycles of extinction and reoccupation of the trait space by interacting species can dilute the expected signal of declining evolutionary rates with the accumulation of species. Moreover, competition is expected to leave a signal of constant or increasing evolutionary rates, as species experience rapid evolutionary rates as they evolve to occupy the edges of the morphospace [17,27].

While a signal of positive diversity-dependence coincides conceptually with a signature of competitive interactions, alternative processes can also generate this pattern. For example, high rates of phenotypic change can be the result of elevated fixation rates, which can be facilitated in small populations during frequent allopatric speciation (i.e. prior to secondary contact). It is also worth mentioning that if species' traits are diverging under competition, and this process further fuels speciation [51], a signal of positive diversity-dependence will emerge, but the underlying mechanism is not the hypothesized increased trait divergence as a consequence of accumulating competitors. Furthermore, our simulated data show that not considering ecoguild membership of clade members can cause high false positives in the signal of diversity-dependence, arguing for caution when making inferences about potential mechanisms underlying such patterns. Estimating the ecoguild membership of species prior to fitting evolutionary models greatly alleviates false positives, although small clades can still show inflated type 1 errors. Out of 59 orders and super-families, five groups that are under 30 species show a strong signal for positive diversity-dependence, and so we could overestimate the prevalence of competitive interactions in these smaller clades. An obvious solution to decrease concerns related to biased model preference would be performing model adequacy tests [42]. Trait evolutionary models with competition are not yet integrated in current model adequacy frameworks (e.g. [42], a framework available for the BM, OU and early burst models only), and thus, it was not possible for us to address this issue here.

We detected a signal for competition in beak shape or relative size in several textbook examples of beak specialization such as flowerpiercers [52], cockatoos [53], vangas and allies [54], sunbirds [55], select kingfishers and bee-eaters [56]. We rarely find a signature of competition in patterns of body mass divergence (but see results for antpittas, smaller radiations within the hummingbirds, tyrant-flycatchers and cotingas, leaf-warblers, bush-warblers and long-tailed tits, chats and Old World flycatchers, and additionally swans and geese, whistlers, and cisticolas in biogeographical analyses). These results are consistent with the tighter link between beak attributes and resource acquisition than with body size [23,54], and also indicate that if small changes in the beak produce substantial differences in the feeding ecology of species, beak change might represent the most parsimonious route towards ecological differentiation [5,10,57]. Conversely, body mass is associated with many aspects of species' biology and it is influenced by many selection forces, and thus, we can expect biotic interactions to play a much smaller relative contribution to body mass evolution

across the bird tree. Overall, these findings provide broad-scale evidence that ecological specialization under competition is usually achieved via one, probably the most parsimonious, route [58]. Our results also indicate that generally the parsimonious route represents considerable divergence in one trait (as opposed to small changes across many), and further argue that a potential signature of competition in the phenotype of extant related species can be missed if studies are restricted to single traits (e.g. [38]). We note that our models could underestimate the signal for competition if species interactions are resolved via divergence in traits not related to beak shape, size or body mass, and when morphology and function are decoupled (e.g. the beak changes shape but maintains the same lever mechanical advantage, or vice versa). Also, competition could be solved via morphological change associated with ecoguild membership switches; in this case, a signature of competition could be missed when only considering trait- and diversity-dependence within ecoguilds.

We find that phylogenetic relatedness is a poor predictor of whether specific groups are prone to competition, in agreement with studies linking the strength of species interactions to stochastic events, such as climatic changes and the associated fluctuations in resource availability [5], or invasion episodes [59]. Furthermore, we expected a higher identifiability and support for competition in more finely divided radiations rather than orders and super-families, because younger and smaller radiations are more likely to include ecologically similar species that will compete over shared resources compared with larger clades, and moreover, even if competition is prevalent, its signature could be erased in deep-time in the large clades [60,61]. Our results suggest that the age of radiations is probably not related to our ability to detect a signal of competition, and incorporating ecoguild reconstructions potentially decreased the loss in identifiability of competition signal with time. We note, however, that age might affect the signature of competition much more among more terminal parts of the phylogeny, a pattern that would not be captured by our correlations between clade age and competition signal.

To conclude, here we estimate the prevalence of patterns consistent with expectations under competitive selection pressures between related lineages (diversity-dependent and phenotypic evolution mediated by similarity in traits) across avian orders and super-families. We find that the signal for competition varies across the avian tree, and methods that assume an effect of species interactions on the evolution of phenotypes are the best description of the data in several putative examples of competition-driven diversification. Overall though, we find that patterns of ecomorphological divergence are generally best captured by models that assume species evolve independently from each other, and that biotic interactions between close relatives leave strong signatures in approximately a third of bird clades (but 45% when considering both ecoguild and biogeographical analyses). Our results indicate that differentiation in one axis of phenotypic divergence is generally sufficient to minimize competition for shared resources. Furthermore, we find that patterns of positive diversity-dependence are more common than trait-dependent or negative diversity-dependent evolution. Lastly, we find that the prevalence of competition signatures is not predicted by phylogenetic relatedness, or age of radiations. Taken together, our results suggest that incorporating competitive selection pressures among clade members into trait

evolutionary models can improve model fit, and that competition from related lineages shapes patterns of biodiversity accumulation beyond a few iconic example groups. However, the potential effects of biotic interactions with closely related species are probably eroded in deep-time, and the ecomorphological diversity of extant species across broad evolutionary scales conserves a mosaic of footprints of multiple selection pressures.

Competing interests. We declare we have no competing interests.

Funding. This work was supported by the European Research Council (grant no. 615709 Project 'ToLERates'), by a Royal Society University Research Fellowship to G.H.T. (UF120016 and URF\R\180006) and by a Leverhulme Early Career Fellowship (ECF-2018-101) to C.R.C.

Acknowledgements. We thank Frane Babarovic, Joseph Brown, Yichen He, Jonathan Kennedy, Thomas Guillerme and Alex Slavenko for valuable comments and discussion points. We thank M. Adams, H. van Grouw and R. Prys-Jones at the Natural History Museum, and H. McGhie at the Manchester Museum for providing access to and expertise in the collections. We thank all the volunteer citizen scientists at http://www.markmybird.org for helping us build the detailed beak shape dataset.

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
