## [Reviewer comments · Proceedings of the Royal Society B: Biological Sciences]

Review History

RSPB-2020-1585.R0 (Original submission)

Review form: Reviewer 1

Recommendation

Accept with minor revision (please list in comments)

Scientific importance: Is the manuscript an original and important contribution to its field?

Excellent

General interest: Is the paper of sufficient general interest?

Excellent

Quality of the paper: Is the overall quality of the paper suitable?

Excellent

Is the length of the paper justified?

Yes

Should the paper be seen by a specialist statistical reviewer?

No

Do you have any concerns about statistical analyses in this paper? If so, please specify them explicitly in your report.

No

It is a condition of publication that authors make their supporting data, code and materials available - either as supplementary material or hosted in an external repository. Please rate, if applicable, the supporting data on the following criteria.

Is it accessible?

Yes

Is it clear?

Yes

Is it adequate?

No

Do you have any ethical concerns with this paper?

No

Comments to the Author

In this manuscript, the authors conduct an impressively large-scale analysis of patterns of trait evolution across all birds, using a massive morphometric dataset compiled from 3D-scans of bird beaks. They find that, although less common than support for models of independent evolution, there is fairly widespread support for models that incorporate competition, particularly in the form of positive diversity dependence, though generally, this support is restricted to one or few traits within each group. They make a strong case for the robustness of their results with several supplementary analyses and a thorough simulation study. Overall, I think this is a very well written and important paper that will contribute to a growing field aiming to bridge the gap between ecological and evolutionary approaches.

The main thing that I think is missing from the paper is a discussion of sympatry. Although the authors build ecological realism into their models by restricting interactions to ecologically similar species ("ecoguilds"), they ignore the biogeographical context of these interactions, effectively assuming that allopatric species can interact with one another. I don't think this is a problem, necessarily, as I recognize the enormous strides that this paper makes as it stands, but I think the potential impacts of this assumption merit a discussion. I suspect that the assumption makes their approach conservative (and their results about species interactions all the more impressive).

Minor comments:

lines 197-200: If I'm understanding this correctly, a different set of models was fit for each of the ecoguilds (i.e., using subgroup pruning as in Drury et al. 2018 PLoS Biology). If this is the case, how were results combined across ecoguilds? Across stochastic maps? If instead, the ecoguilds were treated akin to biogeography (where, in the same model fit, lineages in each ecoguild could interact with one another as if in sympatry, but not with members of other ecoguilds), it wouldn't be accurate to state that "the remaining members of the clade evolve according to a BM model of evolution" because they are themselves impacted by other members of their own ecoguild.

line 201: A bit more information about sampling would be useful here. What was the range of clade sizes? What about the minimum and maximum proportion of lineages in each clade that have morphological measurements?

line 212: Reference 51 seems out of place here, given that Louca & Pennell is about lineage diversification models rather than models of phenotypic evolution (though I see the point of

including it--perhaps the sentence could be reframed about phylogenetic studies based on neontological data only).

lines 314-319: Consider adding a reference here to the figures/tables with this result.

lines 334-338: Did these model fits also generate estimates of phylogenetic signal (i.e., lambda)? If so, it would be worth including them in Tables S3 & S4.

line 385: Are the clades in this parenthetical examples of ones with negative slope estimates? I assume so, but this is a bit unclear as written.

line 405: Is it possible to determine whether measurement error impacts model support and/or parameter estimates from these datasets? Are there data for multiple individuals of each species that could be brought to bear on this question?

line 483: This idea that the effect of biotic interactions erodes through time echos a statement made in the abstract as well (lines 37-40), but I'm not sure I follow the basis for this idea. To me, this would be supported if clade age emerged as a predictor of support for species interactions models, but I might have missed something.

Review form: Reviewer 2

Recommendation

Accept with minor revision (please list in comments)

Scientific importance: Is the manuscript an original and important contribution to its field?

Excellent

General interest: Is the paper of sufficient general interest?

Excellent

Quality of the paper: Is the overall quality of the paper suitable?

Excellent

Is the length of the paper justified?

Yes

Should the paper be seen by a specialist statistical reviewer?

No

Do you have any concerns about statistical analyses in this paper? If so, please specify them explicitly in your report.

No

It is a condition of publication that authors make their supporting data, code and materials available - either as supplementary material or hosted in an external repository. Please rate, if applicable, the supporting data on the following criteria.

Is it accessible?

Yes

Is it clear?

Yes

Is it adequate?

Yes

Do you have any ethical concerns with this paper?

No

Comments to the Author

This is a really interesting and thorough study that provides a comprehensive phylogenetic test of how species interactions may shape phenotypic diversity over macroevolutionary timescales. The authors use an impressive database of measurements on the size and shape of the avian beak (including >8000 species) along with a global bird phylogeny, and compare different models of trait evolution that either include or ignore species interactions. Dividing their phylogeny into multiple clades they find that 1) evolutionary models that include the effects of species interactions are relatively rarely supported but that 2) where the effects of species interactions are evident, rates of trait evolution tend to increase with species diversity. This is an interesting result. It rejects the widely held view that competition and niche filling lead to a slow down in rates of trait evolution over time, but is consistent with the predictions from recent theoretical models suggesting that rates of trait evolution may speed up when competition is intense.

I found the paper to be very clearly written and well presented. The data and modelling approaches are appropriate for the question and the authors are careful to conduct additional simulations and tests to check the robustness of their results to possible model biases. Most of my comments relate to details of the methods where I felt further explanation of discussion of possible caveats is required.

The authors test a diversity dependent model in which the rate of trait evolution varies over time according to the number of lineages in a clade. This is based on the number of lineages leading to extant species and does not include lineages which are now extinct but that may have affected the patterns of trait evolution. I think this is worth pointing out for the general reader and adding a sentence or two in the Discussion about this would be helpful. A related thought, was that although here the authors consider how diversity may influence rates of trait evolution, I was wondering what would happen if causality was reversed: if trait divergence promotes speciation could this also lead to support for a positive diversity-dependent model of trait evolution? Addressing these issues is of course beyond the scope of the present study but it would be interesting if the authors could comment on this.

The authors make the good point that diversity dependence in rates of trait evolution should only apply to species within an 'ecoguild' (i.e. species utilising similar resources and foraging niches). To address this, the authors divide species into four ecoguilds by performing a PCA on diet and foraging data obtained from the Elton Traits database, retaining the first two PC dimensions and finally dividing species into quadrants. They then repeat their diversity dependence models but only allowing species within an ecoguild to interact.

The approach makes sense but the division into quadrants seems a little arbitrary to me. Species could be ecologically very similar but occur in different quadrants, whereas other species could be ecologically very different but still occur in the same quadrant. Because of this, I was wondering what the rationale was for not using the original diet and foraging categories in Elton traits to directly define whether species are within the same ecoguild? I think a few sentences explaining this would be helpful as I'm sure future studies would be interested in applying this kind of approach given the growing availability of ecological and trait datasets.

One point that I think requires discussion is that species may occupy different ecoguilds because of competition (e.g. species feeding at different heights in the forest). Indeed, changes in microhabitat use may be a more common response to competition than changes in morphology. If switching between ecoguilds is a result of competition, then could this lead to the effects of

competition being underestimated when only considering diversity dependence within ecoguilds? This could be especially important if switches between ecoguilds are associated with differences in the morphological traits.

So overall, while I really like the ecoguild analysis, I think there are a number of steps that need some clearer explanation and discussion.

In relation to the above analysis, throughout the paper it was not clear to me what was meant by “structured covariances among species’ phenotypes” so I think this needs some unpacking.

While the authors account for differences among species in guild membership they do not consider the effects of geography: species can only compete if they live in the same place. I imagine that matrices of geographic co-occurrence could be included in the same way as the authors do for ecoguilds. Is this something the authors have explored? I think this should at least be raised as a possible explanation for the patterns observed (i.e. why a more consistent signature of competition is not found across clades).

The finding that clades tend to exhibit evidence of competition along just one and not multiple trait axes is really interesting. However, I was wondering whether this is simply what you would expect given the low prevalence of support for these models across clades. Perhaps the authors could perform a randomisation to see if the number of times when you get support for competition driving trait evolution in one versus multiple trait axes is different from what you would expect by chance.

L33: can you give more detail here on the clades, just to give an idea of the phylogenetic scales you are examining

L83: Rather than ‘favours competition’, perhaps ‘corresponds to stronger competition’

L99: the diversity dependent model considers both positive and negative diversity dependence.

Does the trait-dependent model also include the possibility that species converge towards the mean rather than differentiate? Would this end up being very similar to an OU model?

L201: Helpful to know why it wasn’t computationally possible.

L218: citation for this?

Decision letter (RSPB-2020-1585.R0)

07-Aug-2020

Dear Dr Chira:

Your manuscript has now been peer reviewed and the reviews have been assessed by an Associate Editor. The reviewers’ comments (not including confidential comments to the Editor) and the comments from the Associate Editor are included at the end of this email for your reference. As you will see, the reviewers and the Editors have raised some concerns with your manuscript and we would like to invite you to revise your manuscript to address them.

To submit your revision please log into <http://mc.manuscriptcentral.com/prsb> and enter your Author Centre, where you will find your manuscript title listed under "Manuscripts with

Decisions." Under "Actions", click on "Create a Revision". Your manuscript number has been appended to denote a revision.

Research ethics:

Use of animals and field studies:

It is a condition of publication that you make available the data and research materials supporting the results in the article. Please see our Data Sharing Policies (<https://royalsociety.org/journals/authors/author-guidelines/#data>). Datasets should be deposited in an appropriate publicly available repository and details of the associated accession number, link or DOI to the datasets must be included in the Data Accessibility section of the article (<https://royalsociety.org/journals/ethics-policies/data-sharing-mining/>). Reference(s) to datasets should also be included in the reference list of the article with DOIs (where available).

All supplementary materials accompanying an accepted article will be treated as in their final form. They will be published alongside the paper on the journal website and posted on the online figshare repository. Files on figshare will be made available approximately one week before the

accompanying article so that the supplementary material can be attributed a unique DOI. Please try to submit all supplementary material as a single file.

Please submit a copy of your revised paper within three weeks. If we do not hear from you within this time your manuscript will be rejected. If you are unable to meet this deadline please let us know as soon as possible, as we may be able to grant a short extension.

Best wishes,
Dr Daniel Costa
mailto:proceedingsb@royalsociety.org

Associate Editor

Board Member: 1

Comments to Author:

This is a very nice paper that uses an impressive avian dataset to investigate the potential association between interspecific competition and phenotypic evolution. It is well written, and the results are interesting and of broad interest. Although one important point raised by both reviewers would require a simple new re-analysis of the data, overall I agree with both reviewers that it mostly requires minor adjustments before it is ready for publication. Additional to all the good suggestions provided by both reviews I would add (and in fact reinforce one point raised by both reviewers) the following suggestions/points:

1) The authors correctly point out that it is necessary to take into account the ecological guild identity of the species when fitting diversity dependent and trait competition models, both from a conceptual as well as a statistical point of view. In line with the conceptual justification, both reviewers pointed out that spatial coexistence is another relevant aspect that was not considered. I fully agree with them and I think this is a very important aspect that should be addressed. I understand finding the “correct” spatial scale might not be an easy task, but I think that considering it at least at a broad scale (e.g. being at the same continent) would in fact improve the analysis and argument of the paper. Incorporating space might even reveal a stronger role for competition. Depending on how “sympatry” is considered, it would not require a new type of analysis but rather to simply add another “trait” to the guild categorization, which would be a “ecological guild + spatial overlap” characterization.

2) I wasn't sure why it is not possible to do some kind of model adequacy (lines 422-424). Couldn't the process be simulated using the current simulation tools and then some empirical characteristics (e.g. distribution of trait values) compared to the simulated distribution? There might not be a “clean” way of doing it but maybe something.... I am not familiar with those phenotypic models and I might have missed something here, so sorry if I did. Alternatively, it might be interesting to explain a bit more in the text those difficulties, maybe in a supplemental material if it disrupts the narrative.

3) If I understood it correctly all the analyses were done using an MCC tree. Why not take phylogenetic uncertainty into account and use a distribution of trees? Would the computational time be prohibitive? I know this would not be ideal, but maybe using just 10 trees would be interesting in this respect.

Reviewer(s)' Comments to Author:

Referee: 1

Comments to the Author(s)

In this manuscript, the authors conduct an impressively large-scale analysis of patterns of trait evolution across all birds, using a massive morphometric dataset compiled from 3D-scans of bird beaks. They find that, although less common than support for models of independent evolution, there is fairly widespread support for models that incorporate competition, particularly in the form of positive diversity dependence, though generally, this support is restricted to one or few traits within each group. They make a strong case for the robustness of their results with several supplementary analyses and a thorough simulation study. Overall, I think this is a very well written and important paper that will contribute to a growing field aiming to bridge the gap between ecological and evolutionary approaches.

The main thing that I think is missing from the paper is a discussion of sympatry. Although the authors build ecological realism into their models by restricting interactions to ecologically similar species ("ecoguilds"), they ignore the biogeographical context of these interactions, effectively assuming that allopatric species can interact with one another. I don't think this is a problem, necessarily, as I recognize the enormous strides that this paper makes as it stands, but I think the potential impacts of this assumption merit a discussion. I suspect that the assumption makes their approach conservative (and their results about species interactions all the more impressive).

Minor comments:

lines 197-200: If I'm understanding this correctly, a different set of models was fit for each of the ecoguilds (i.e., using subgroup pruning as in Drury et al. 2018 PLoS Biology). If this is the case, how were results combined across ecoguilds? Across stochastic maps? If instead, the ecoguilds were treated akin to biogeography (where, in the same model fit, lineages in each ecoguild could interact with one another as if in sympatry, but not with members of other ecoguilds), it wouldn't be accurate to state that "the remaining members of the clade evolve according to a BM model of evolution" because they are themselves impacted by other members of their own ecoguild.

line 201: A bit more information about sampling would be useful here. What was the range of clade sizes? What about the minimum and maximum proportion of lineages in each clade that have morphological measurements?

line 212: Reference 51 seems out of place here, given that Louca & Pennell is about lineage diversification models rather than models of phenotypic evolution (though I see the point of including it--perhaps the sentence could be reframed about phylogenetic studies based on neontological data only).

lines 314-319: Consider adding a reference here to the figures/tables with this result.

lines 334-338: Did these model fits also generate estimates of phylogenetic signal (i.e., lambda)? If so, it would be worth including them in Tables S3 & S4.

line 385: Are the clades in this parenthetical examples of ones with negative slope estimates? I assume so, but this is a bit unclear as written.

line 405: Is it possible to determine whether measurement error impacts model support and/or parameter estimates from these datasets? Are there data for multiple individuals of each species that could be brought to bear on this question?

line 483: This idea that the effect of biotic interactions erodes through time echos a statement made in the abstract as well (lines 37-40), but I'm not sure I follow the basis for this idea. To me, this would be supported if clade age emerged as a predictor of support for species interactions models, but I might have missed something.

Referee: 2

Comments to the Author(s)

This is a really interesting and thorough study that provides a comprehensive phylogenetic test of how species interactions may shape phenotypic diversity over macroevolutionary timescales. The authors use an impressive database of measurements on the size and shape of the avian beak (including >8000 species) along with a global bird phylogeny, and compare different models of trait evolution that either include or ignore species interactions. Dividing their phylogeny into multiple clades they find that 1) evolutionary models that include the effects of species interactions are relatively rarely supported but that 2) where the effects of species interactions are evident, rates of trait evolution tend to increase with species diversity. This is an interesting result. It rejects the widely held view that competition and niche filling lead to a slow down in rates of trait evolution over time, but is consistent with the predictions from recent theoretical models suggesting that rates of trait evolution may speed up when competition is intense.

I found the paper to be very clearly written and well presented. The data and modelling approaches are appropriate for the question and the authors are careful to conduct additional simulations and tests to check the robustness of their results to possible model biases. Most of my comments relate to details of the methods where I felt further explanation of discussion of possible caveats is required.

The authors test a diversity dependent model in which the rate of trait evolution varies over time according to the number of lineages in a clade. This is based on the number of lineages leading to extant species and does not include lineages which are now extinct but that may have affected the patterns of trait evolution. I think this is worth pointing out for the general reader and adding a sentence or two in the Discussion about this would be helpful. A related thought, was that although here the authors consider how diversity may influence rates of trait evolution, I was wondering what would happen if causality was reversed: if trait divergence promotes speciation could this also lead to support for a positive diversity-dependent model of trait evolution? Addressing these issues is of course beyond the scope of the present study but it would be interesting if the authors could comment on this.

The authors make the good point that diversity dependence in rates of trait evolution should only apply to species within an 'ecoguild' (i.e. species utilising similar resources and foraging niches). To address this, the authors divide species into four ecoguilds by performing a PCA on diet and foraging data obtained from the Elton Traits database, retaining the first two PC dimensions and finally dividing species into quadrants. They then repeat their diversity dependence models but only allowing species within an ecoguild to interact.

The approach makes sense but the division into quadrants seems a little arbitrary to me. Species could be ecologically very similar but occur in different quadrants, whereas other species could be ecologically very different but still occur in the same quadrant. Because of this, I was wondering what the rationale was for not using the original diet and foraging categories in Elton traits to directly define whether species are within the same ecoguild? I think a few sentences explaining this would be helpful as I'm sure future studies would be interested in applying this kind of approach given the growing availability of ecological and trait datasets.

One point that I think requires discussion is that species may occupy different ecoguilds because of competition (e.g. species feeding at different heights in the forest). Indeed, changes in microhabitat use may be a more common response to competition than changes in morphology.

If switching between ecoguilds is a result of competition, then could this lead to the effects of competition being underestimated when only considering diversity dependence within ecoguilds? This could be especially important if switches between ecoguilds are associated with differences in the morphological traits.

So overall, while I really like the ecoguild analysis, I think there are a number of steps that need some clearer explanation and discussion.

In relation to the above analysis, throughout the paper it was not clear to me what was meant by “structured covariances among species’ phenotypes” so I think this needs some unpacking.

While the authors account for differences among species in guild membership they do not consider the effects of geography: species can only compete if they live in the same place. I imagine that matrices of geographic co-occurrence could be included in the same way as the authors do for ecoguilds. Is this something the authors have explored? I think this should at least be raised as a possible explanation for the patterns observed (i.e. why a more consistent signature of competition is not found across clades).

The finding that clades tend to exhibit evidence of competition along just one and not multiple trait axes is really interesting. However, I was wondering whether this is simply what you would expect given the low prevalence of support for these models across clades. Perhaps the authors could perform a randomisation to see if the number of times when you get support for competition driving trait evolution in one versus multiple trait axes is different from what you would expect by chance.

L33: can you give more detail here on the clades, just to give an idea of the phylogenetic scales you are examining

L83: Rather than ‘favours competition’, perhaps ‘corresponds to stronger competition’

L99: the diversity dependent model considers both positive and negative diversity dependence. Does the trait-dependent model also include the possibility that species converge towards the mean rather than differentiate? Would this end up being very similar to an OU model?

L201: Helpful to know why it wasn’t computationally possible.

L218: citation for this?

Author's Response to Decision Letter for (RSPB-2020-1585.R0)

See Appendix A.

RSPB-2020-1585.R1 (Revision)

Review form: Reviewer 1

Recommendation

Accept as is

Scientific importance: Is the manuscript an original and important contribution to its field?

Excellent

General interest: Is the paper of sufficient general interest?

Excellent

Quality of the paper: Is the overall quality of the paper suitable?

Excellent

Is the length of the paper justified?

Yes

Should the paper be seen by a specialist statistical reviewer?

No

Do you have any concerns about statistical analyses in this paper? If so, please specify them explicitly in your report.

No

It is a condition of publication that authors make their supporting data, code and materials available - either as supplementary material or hosted in an external repository. Please rate, if applicable, the supporting data on the following criteria.

Is it accessible?

Yes

Is it clear?

Yes

Is it adequate?

Yes

Do you have any ethical concerns with this paper?

No

Comments to the Author

The authors have done an excellent job revising their manuscript (I was reviewer 1 before), and I have no further comments or suggestions. I look forward to seeing the paper in print.

Review form: Reviewer 2

Recommendation

Accept as is

Scientific importance: Is the manuscript an original and important contribution to its field?

Excellent

General interest: Is the paper of sufficient general interest?

Excellent

Quality of the paper: Is the overall quality of the paper suitable?

Excellent

Is the length of the paper justified?

Yes

Should the paper be seen by a specialist statistical reviewer?

No

Do you have any concerns about statistical analyses in this paper? If so, please specify them explicitly in your report.

No

It is a condition of publication that authors make their supporting data, code and materials available - either as supplementary material or hosted in an external repository. Please rate, if applicable, the supporting data on the following criteria.

Is it accessible?

Yes

Is it clear?

Yes

Is it adequate?

Yes

Do you have any ethical concerns with this paper?

No

Comments to the Author

I noticed just one typo in the new version of the paper
L298 delete 'show'

Decision letter (RSPB-2020-1585.R1)

16-Oct-2020

Dear Dr Chira

I am pleased to inform you that your manuscript entitled "The signature of competition in ecomorphological traits across the avian radiation" has been accepted for publication in Proceedings B.

Open Access

Paper charges

Sincerely,

Dr Daniel Costa

Associate Editor:

Board Member: 1

Comments to Author:

I agree with the reviewers that the authors have done a great job revising the manuscript. Given the inherent limitations to such kind of analysis, the authors have done a great job addressing previous comments. This is a very interesting paper. There are not further comments except one typo pointed out by reviewer 2.

Appendix A

Department of Animal and Plant Sciences
University of Sheffield
Sheffield, S10 2TN

Email: angelamchira@gmail.com

The
University
Of
Sheffield.

To the Editors,

We hereby submit the revised version of the manuscript entitled “The signature of competition in ecomorphological traits across the avian radiation” (RSPB-2020-1585) for consideration for publication as a research article in *Proceedings B*. We have included the editor and reviewer’s comments below, followed by our response to each of these comments.

Your sincerely,
Angela Chira (on behalf of all co-authors)

Statement

The current version of the manuscript has been seen and approved by all authors, and further, it has not been submitted for consideration elsewhere.

Associate Editor, Board Member:1
Comments to Author:

This is a very nice paper that uses an impressive avian dataset to investigate the potential association between interspecific competition and phenotypic evolution. It is well written, and the results are interesting and of broad interest. Although one important point raised by both reviewers would require a simple new re-analysis of the data, overall I agree with both reviewers that it mostly requires minor adjustments before it is ready for publication. Additional to all the good suggestions provided by both reviews I would add (and in fact reinforce one point raised by both reviewers) the following suggestions/points:

1) The authors correctly point out that it is necessary to take into account the ecological guild identity of the species when fitting diversity dependent and trait competition models, both from a conceptual as well as a statistical point of view. In line with the conceptual justification, both reviewers pointed out that spatial coexistence is another relevant aspect that was not considered. I fully agree with them and I think this is a very important aspect that should to be addressed. I understand finding the “correct” spatial scale might not be an easy task, but I think that considering it at least at a broad scale (e.g. being at the same continent) would in fact improve the analysis and argument of the paper. Incorporating space might even reveal a stronger role for competition. Depending on how “sympatry” is considered, it would not require a new type of analysis but rather to simply

add another “trait” to the guild categorization, which would be a “ecological guild + spatial overlap” characterization.

- We agree that the biogeographical context of species interactions is very important. We address the editor and the reviewers’ comments by including a biogeography set of analyses in the revisions. Briefly: for each clade, we use breeding range maps provided by Birdlife to build a presence-absence matrix. From this matrix, we build a matrix of co-existence indices for all pairs of species within the clade (1 if species’ ranges overlap, and 0 if they do not). We further use the clustering algorithm implemented in the function `upgma()` in the R package *phangorn* to identify sets of species that broadly overlap in spatial distribution. The function takes a distance matrix; in our case, the distance between any two species is represented by 1 - their co-existence index. This way, a pair of species is separated by a distance of 0 if they co-exist somewhere in their range, and a distance of 1 if their ranges do not overlap. Each cluster of species is given a geographic cluster index. We use these cluster indices as equivalents of discrete characters and use `make.simmap()` in the R package *phytools* to build stochastic maps that estimate the biogeography history of clades. While biogeography reconstructions can be implemented in this way in models with competition, such an approach is recommended for clades with simple biogeography relations between species (e.g. the radiation of *Anolis* lizards on a small number of islands, and in which many species are restricted to single islands). In our analyses, we exclude clusters with less than five species, in order to avoid cases with large numbers of clusters per clade that could lead to unreliable biogeography reconstructions. Further, combining the geography and ecoguild structure can also lead to a very high number of clusters per clade, and so we perform these analyses separate from the ecoguild analyses. The biogeography analyses show similar patterns to the main analyses. Expectedly, in some clades, purely geographic analyses miss a signal for competition otherwise revealed by taking into account ecoguild membership. However, these analyses also uncover a signal for competition in several clades where it was previously missed. Although crude, this approach encourages the exploration of more sophisticated ways of combining both geographic and ecological relationships between species. These analyses are presented in detail in the supplementary material (supplementary information on the quantification of biogeographic reconstructions, Table S1b, Figure S2d, Figure S3), and discussed in the main text in lines 222-231, 294-301, 366-367.

2) I wasn’t sure why it is not possible to do some kind of model adequacy (lines 422-424). Couldn’t the process be simulated using the current simulation tools and then some empirical characteristics (e.g. distribution of trait values) compared to the simulated distribution? There might not be a “clean” way of doing it but maybe something.... I am not familiar with those phenotypic models and I might have missed something here, so sorry if I did. Alternatively, it might be interesting to explain a bit more in the text those difficulties, maybe in a supplemental material if it disrupts the narrative.

- We have clarified in the text that we refer to the model adequacy framework proposed by Pennell et al 2015, an approach available for the BM, OU and EB models (lines 433-434). While we could simulate trait distributions under the

current models, finding key statistics that would determine the adequacy of models with competition would not be trivial, and further, analyzing the simulated data would be computationally very expensive. Unfortunately, we are forced to leave this matter for further studies.

3) If I understood it correctly all the analyses were done using an MCC tree. Why not take phylogenetic uncertainty into account and use a distribution of trees? Would the computational time be prohibitive? I know this would not be ideal, but maybe using just 10 trees would be interesting in this respect.

- We agree that using a distribution of trees from the posterior would be ideal to deal with phylogenetic uncertainty, and we have considered adding such an analysis in the revision. However, it quickly became apparent that such an approach is computationally prohibitive, even for a small number of trees. The empirical data has a range of ecoguild structures and tree sizes, as opposed to the simulations we performed to evaluate type 1 errors. Controlling for phylogenetic uncertainty using the posterior distribution of trees generally requires running models over hundreds of trees to make sure the observed patterns are not the result of odd trees from the posterior (the MCC tree in these analyses is built over 10,000 posterior trees). We do however include analyses using trees built only with species for which genetic data is available. These trees should be less prone to phylogenetic error, and the consistency of observed patterns over genetic vs full trees is an indicator for the robustness of the observed patterns. Further, we also implement a new check for bias in model inference due to phylogenetic uncertainty: for each tree, we use `treedist()` to calculate four metrics of tree distances between the MCC tree and all the trees in the posterior. These metrics are an estimation of how much the posterior distribution differs from the MCC tree. We use the average tree distance across the posterior (as well as average of the 75% quantile in order to get a metric of the more extreme cases) as a measure of phylogenetic uncertainty for each clade, i.e. a metric of dissimilarity between the consensus tree and the posterior. We find no correlation between the prevalence of competition signal and this measure of phylogenetic uncertainty, which supports the robustness of the results to high levels of differences between the MCC tree and the posterior distribution of trees. The new analyses are explained in the supplementary material (supplementary information on the quantification of phylogenetic uncertainty estimates using the posterior distribution of trees, Table S8), as well as in the main text (lines 214-220, 354-356).

Reviewer(s)' Comments to Author:

Referee: 1

Comments to the Author(s)

In this manuscript, the authors conduct an impressively large-scale analysis of patterns of trait evolution across all birds, using a massive morphometric dataset compiled from 3D-scans of bird beaks. They find that, although less common than support for models of independent evolution, there is fairly widespread support for models that incorporate competition, particularly in the form of positive diversity dependence, though generally, this support is restricted to one or few traits within each group. They make a strong case for the robustness of their results with several supplementary analyses and a thorough simulation study. Overall, I think this is a very well written and important paper that will contribute to a growing field aiming to bridge the gap between ecological and evolutionary approaches.

The main thing that I think is missing from the paper is a discussion of sympatry. Although the authors build ecological realism into their models by restricting interactions to ecologically similar species ("ecoguilds"), they ignore the biogeographical context of these interactions, effectively assuming that allopatric species can interact with one another. I don't think this is a problem, necessarily, as I recognize the enormous strides that this paper makes as it stands, but I think the potential impacts of this assumption merit a discussion. I suspect that the assumption makes their approach conservative (and their results about species interactions all the more impressive).

- Please see our comments above on the new biogeography analyses added.

Minor comments:

lines 197-200: If I'm understanding this correctly, a different set of models was fit for each of the ecoguilds (i.e., using subgroup pruning as in Drury et al. 2018 PLoS Biology). If this is the case, how were results combined across ecoguilds? Across stochastic maps? If instead, the ecoguilds were treated akin to biogeography (where, in the same model fit, lineages in each ecoguild could interact with one another as if in sympatry, but not with members of other ecoguilds), it wouldn't be accurate to state that "the remaining members of the clade evolve according to a BM model of evolution" because they are themselves impacted by other members of their own ecoguild.

- This is a very good point. The former explanation was referring to a dummy example in which some species would be part of the same ecoguild (and thus interact), while, if the remaining would be all part of different ecoguilds, they would each evolve according to a BM model. We agree that this explanation was confusing and have clarified the text in lines 199-200.

line 201: A bit more information about sampling would be useful here. What was the range of clade sizes? What about the minimum and maximum proportion of lineages in each clade that have morphological measurements?

- We have added the range of clade sizes, as well as the range for the percentage of species with morphological data in each clade (lines 153-157).

line 212: Reference 51 seems out of place here, given that Louca & Pennell is about lineage diversification models rather than models of phenotypic evolution (though I see the point of including it--perhaps the sentence could be reframed about phylogenetic studies based on neontological data only).

- We agree that the study of Louca and Pennell concerns lineage diversification processes. We had included it here as an extension for the raised issue of multiple possible scenarios (here given a distribution of tip values). However, for maximum clarity, we have excluded it from the suite of references at this point.

lines 314-319: Consider adding a reference here to the figures/tables with this result.

- We have added the according references to Figures S5-S7.

lines 334-338: Did these model fits also generate estimates of phylogenetic signal (i.e., λ)? If so, it would be worth including them in Tables S3 & S4.

- The phylogenetic logistic regression does not return estimates of phylogenetic signal.

line 385: Are the clades in this parenthetical examples of ones with negative slope estimates? I assume so, but this is a bit unclear as written.

- We have modified the text to be more clear that the examples in the parenthesis are indeed clades with negative slope estimates (line 392).

line 405: Is it possible to determine whether measurement error impacts model support and/or parameter estimates from these datasets? Are there data for multiple individuals of each species that could be brought to bear on this question?

- Unfortunately, we do not have estimates of measurement error, as our data comprises of one individual per species. We have now added this note in the text as well (lines 142-143).

line 483: This idea that the effect of biotic interactions erodes through time echos a statement made in the abstract as well (lines 37-40), but I'm not sure I follow the basis for this idea. To me, this would be supported if clade age emerged as a predictor of support for species interactions models, but I might have missed something.

- We agree that the relationship between clade age and a signal for competition would provide quantitative support for the idea of signal erosion with time. There is, however, empirical evidence that competition can affect trait evolution for a very small numbers of species, for example, with the fluctuation of resources (e.g. Darwin's finches, Grant and Grant 2003, 2006). Such signals can, however, be eroded or lost in deep-time, and thus, not detectable in the current distribution of traits (and hence by our models with competition). For this reason, we cannot speculate on the effects of competition over small temporal scales, and we use the more conservative argument of signal erosion in deep time when we refer to an infrequent prevalence of competition. Further, if age affects the signature of

competition on the very terminal parts of phylogenies, such a pattern would not be detected by a relationship between clade age and competition signal. We have now clarified the text in lines 376-377, 471-476 to explain the apparent contradiction between the idea of loss of signal in deep-time and the lack of relationship between clade age and competition signal.

Referee: 2

Comments to the Author(s)

This is a really interesting and thorough study that provides a comprehensive phylogenetic test of how species interactions may shape phenotypic diversity over macroevolutionary timescales. The authors use an impressive database of measurements on the size and shape of the avian beak (including >8000 species) along with a global bird phylogeny, and compare different models of trait evolution that either include or ignore species interactions. Dividing their phylogeny into multiple clades they find that 1) evolutionary models that include the effects of species interactions are relatively rarely supported but that 2) where the effects of species interactions are evident, rates of trait evolution tend to increase with species diversity. This is an interesting result. It rejects the widely held view that competition and niche filling lead to a slow down in rates of trait evolution over time, but is consistent with the predictions from recent theoretical models suggesting that rates of trait evolution may speed up when competition is intense.

I found the paper to be very clearly written and well presented. The data and modelling approaches are appropriate for the question and the authors are careful to conduct additional simulations and tests to check the robustness of their results to possible model biases. Most of my comments relate to details of the methods where I felt further explanation or discussion of possible caveats is required.

The authors test a diversity dependent model in which the rate of trait evolution varies over time according to the number of lineages in a clade. This is based on the number of lineages leading to extant species and does not include lineages which are now extinct but that may have affected the patterns of trait evolution. I think this is worth pointing out for the general reader and adding a sentence or two in the Discussion about this would be helpful. A related thought, was that although here the authors consider how diversity may influence rates of trait evolution, I was wondering what would happen if causality was reversed: if trait divergence promotes speciation could this also lead to support for a positive diversity-dependent model of trait evolution? Addressing these issues is of course beyond the scope of the present study but it would be interesting if the authors could comment on this.

- These are very good points. We have now added a sentence in Discussion about caveats of using extant lineages only (lines 402-406). We also touch on the identifiability of a signal of positive diversity-dependence if speciation itself is promoted by trait divergence in lines 415-422.

The authors make the good point that diversity dependence in rates of trait evolution should only apply to species within an 'ecoguild' (i.e. species utilising similar resources and foraging niches). To address this, the authors divide species into four ecoguilds by performing a PCA on diet and foraging data obtained from the Elton Traits database, retaining the first two PC dimensions and finally dividing species into quadrants. They then repeat their diversity dependence models but only allowing species within an ecoguild to interact.

The approach makes sense but the division into quadrants seems a little arbitrary to me. Species could be ecologically very similar but occur in different quadrants, whereas other species could be ecologically very different but still occur in the same quadrant. Because of this, I was wondering what the rationale was for not using the original diet and foraging categories in Elton traits to directly define whether species are within the same ecoguild? I think a few sentences explaining this would be helpful as I'm sure future studies would be interested in applying this kind of approach given the growing availability of ecological and trait datasets.

- We agree that a PCA on foraging and diet categories is not perfect. We decided to approach ecoguilds via a PCA due to the correlation of foraging and diet metrics, as well as due to the great number of possible categories using the original Elton Traits data. Using quadrants allowed us to divide each clade into four ecoguilds, which made the reconstructions of ecoguild membership (stochastic maps) manageable. Even with four ecoguilds, some stochastic maps resulted in unreliable reconstructions, with several changes in ecoguilds on the same branch. These reconstructions further led to computational issues when building the ecoguild object and/or when running models with competition, which resulted in the exclusion of clades. We have now added a more detailed explanation of this rationale (lines 200-207). We have also modified Figure S1b to show both the ecoguild morphospace, and how the original Elton Trait categories shape the ecoguild space into four quadrants.

One point that I think requires discussion is that species may occupy different ecoguilds because of competition (e.g. species feeding at different heights in the forest). Indeed, changes in microhabitat use may be a more common response to competition than changes in morphology. If switching between ecoguilds is a result of competition, then could this lead to the effects of competition being underestimated when only considering diversity dependence within ecoguilds? This could be especially important if switches between ecoguilds are associated with differences in the morphological traits.

- This is a very good point. We have now added some discussion points on this matter (lines 455-461), addressing the ideas that we could underestimate competition signal if competition is solved by changes that are not associated with beak shape, size, or body mass (diet, foraging strategy, or other types of behavioral activities). Moreover, we note that if changes in focal morphological traits are associated with switches between ecoguilds, the models could underestimate competition signal by when only considering trait-dependence and diversity-dependence within ecoguilds.

So overall, while I really like the ecoguild analysis, I think there are a number of steps that need some clearer explanation and discussion.

In relation to the above analysis, throughout the paper it was not clear to me what was meant by "structured covariances among species' phenotypes" so I think this needs some unpacking.

- We have now expanded the explanation of this term when it first occurs (lines 74-79). Lines 180-184 also expand on the issue.

While the authors account for differences among species in guild membership they do not consider the effects of geography: species can only compete if they live in the same place. I imagine that matrices of geographic co-occurrence could be included in the same way as the authors do for ecoguilds. Is this something the authors have explored? I think this should at least be raised as a possible explanation for the patterns observed (i.e. why a more consistent signature of competition is not found across clades).

- Please see our comments above on the new biogeography analyses added.

The finding that clades tend to exhibit evidence of competition along just one and not multiple trait axes is really interesting. However, I was wondering whether this is simply what you would expect given the low prevalence of support for these models across clades. Perhaps the authors could perform a randomisation to see if the number of times when you get support for competition driving trait evolution in one versus multiple trait axes is different from what you would expect by chance.

- We have performed a statistical randomization test. We build a matrix with 3 columns (beak shape, size, and body mass) and 59 rows (1 row per clade), and we populate it with 0. Given 22/59 clades show competition across beak shape, size, and body mass, we assign 22 values of 1 at random to the 177 (59x3) matrix positions and repeat this step 5000 times. The results show that, across 5000 repetitions, the number of clades that show competition in two or more traits (within a given repetition) ranges from 0 to 7, with a median of 3. Therefore, statistically, given an a priori low prevalence of competition signal, we generally do not expect it to appear across multiple traits in the same clade. This being said, such statistical tests do not inform much about biology. The number of clades that show competition across beak size, shape and body mass is not known a priori, rather it is the sum of the number of clades showing competition in either of these traits. The issue is thus not about assigning an a priori quantity among clades & traits, in which case the statistical randomization would inform on expected frequencies. We can imagine a scenario in which competition is resolved via small changes across multiple axes of diversification, in which case, we would still have a low, but not infrequent, prevalence of competition signal when we consider any of the traits, but these signals would be clustered in the same clades.

L33: can you give more detail here on the clades, just to give an idea of the phylogenetic scales you are examining

- We have added the range of clade sizes in the abstract as well, to help clarify the phylogenetic scales considered.

L83: Rather than 'favours competition', perhaps 'corresponds to stronger competition'

- We have now modified the text (line 86).

L99: the diversity dependent model considers both positive and negative diversity dependence. Does the trait-dependent model also include the possibility that species

converge towards the mean rather than differentiate? Would this end up being very similar to an OU model?

- The current implementation of models with competition does not allow convergence. This issue is discussed in Drury et al 2016 in relation to the matching competition model, where it is stated that the S parameter and the parameter accounting for the pull of species towards the mean are not identifiable. Drury et al 2018 simulate data under both convergence and divergence, and find out that, when applying models of evolution, the OU model is generally picked up as best in that case. We have now added a sentence in discussion to approach these issues (lines 398-402).

L201: Helpful to know why it wasn't computationally possible.

- We have now added in text the explanation related to the fact that ecoguild reconstruction sometimes resulted in patterns of several shifts in character on the same branch, which made the ecoguild object unreliable and resulted in the failure of models with competition in such cases (lines 203-207).

L218: citation for this?

- We have added Drury et al 2016 as an example citation (in the study, Figure 4b shows that when the generating model is OU, exponential diversity-dependent models can be favoured).